



# Continuous measurement of air-water gas exchange by underwater eddy covariance

Peter Berg, Michael. L. Pace

Department of Environmental Sciences, University of Virginia, Charlottesville, Virginia, USA

Correspondence to: Peter Berg (pb8n@virginia.edu)

**Abstract.** Exchange of gasses, such as $O_2$, $CO_2$, and $CH_4$, over the air-water interface is an important component in aquatic ecosystem studies, but exchange rates are typically measured or estimated with substantial uncertainties. This diminishes the precision of common ecosystem assessments associated with gas exchanges such as primary production, respiration, and greenhouse gas emission. Here, we use the aquatic eddy covariance technique – originally

developed for benthic $O_2$ flux measurements – right below the air-water interface ($\sim$5 cm) to determine gas exchange rates and coefficients. Using an Acoustic Doppler Velocimeter and a fast-responding dual $O_2$-temperature sensor mounted on a floating platform, the 3D water velocity, $O_2$ concentration, and temperature are measured at high-speed (64 Hz). By combining these data, concurrent vertical fluxes of $O_2$ and heat across the air-water interface are derived, and from the

former, gas exchange coefficients. Proof-of-concept deployments at different river sites gave standard gas exchange coefficients ($k_{600}$) in the range of published values. A 40 h long deployment revealed a distinct diurnal pattern in air-water exchange of $O_2$ that was controlled largely by physical processes (e.g., diurnal variations in air temperature and associated air-water heat fluxes) and not by biological activity (primary production and respiration). This physical control

of gas exchange is prevalent in lotic systems and adds uncertainty to common ecosystem assessments of biological activity relying on water column $O_2$ concentration recordings. For





example, in the 40 h deployment, there was close-to constant river flow and insignificant winds – two main drivers of lotic gas exchange – but we found gas exchange coefficients that varied by several fold. This was presumably caused by vertical temperature-density gradient formation and

erosion in the surface water driven by the heat flux into or out of the river that controlled the turbulent mixing. This effect is unaccounted for in widely used empirical correlations for gas exchange coefficients and is another source of uncertainty in gas exchange estimates. The aquatic eddy covariance technique allows studies of air-water gas exchange processes and their controls at an unparalleled level of detail. A finding related to the new approach is that heat fluxes at the

air-water interface can, contrary to those typically found in the benthic environment, be substantial and require correction of $O_2$ sensor readings using high-speed parallel temperature measurements. Fast-responding $O_2$ sensors are inherently sensitive to temperature changes, and if this correction is omitted, temperature fluctuations associated with the turbulent heat flux will mistakenly be recorded as $O_2$ fluctuations and bias the $O_2$ eddy flux calculation.


## 1 Introduction

### 1.1 Background

Exchange rates of gasses over the air-water interface in rivers, streams, reservoirs, lakes, and

estuaries are key parameters for estimating a number of important ecosystem variables (Cole et al. 2010). Gas exchange rates are used to estimate metabolism of aquatic systems (Hanson et al. 2004; Van de Bogert et al. 2007; Van de Bogert et al. 2012), emission of greenhouse gasses like $CO_2$ and $CH_4$ to the atmosphere (Cole et al. 2010), and the role of inland and near-shore waters in regional (Billett and Moore 2008) and global (Cole et al. 2007; Bastviken et al. 2011) carbon

cycling. As a result, over several decades a tremendous effort among aquatic scientists has focused on understanding and quantifying gas exchange processes at the air-water interface and their controls under naturally occurring field conditions (Whitman 1923; Butman and Raymond 2011; Raymond et al. 2013).



Multiple state variables and complex physical processes on both sides of the air-water interface
control gas exchange (Macintyre et al. 1995; MacIntyre et al. 2010). Despite this complexity, the
widely used expression for gas exchange rates was formulated based on a conceptually simple
model assuming that gas is transported by molecular diffusion across intact boundary layers, or
thin films, found on each side of the interface (Whitman 1923; Liss and Slater 1974):


$$J_{air-water} = k(C_{water} - C_{air})$$   (1)

where $J_{air-water}$ is the exchange rate, or vertical flux, of the gas (positive upward), $C_{water}$ is the gas
bulk concentration below the film on the water-side, $C_{air}$ is the concentration above the film on the
air-side, and $k$ is the gas exchange coefficient, often also referred to as the 'gas transfer velocity' or
'piston velocity'. For most gasses, $C_{water}$ and $C_{air}$ are straight forward to measure with modern
sensors (Koopmans and Berg 2015; Fritzsche et al. 2017), or calculate from known functions, but
the complexity of gas exchange and its many controlling variables is contained in $k$ (Macintyre et
al. 1995; McKenna and McGillis 2004; Cole et al. 2010).


For sparingly soluble gasses such as $O_2$, $CO_2$, and $CH_4$, the ratio between the molecular diffusivity
in air and water is on the order of $10^4$. Consequently, the resistance to gas diffusion is associated
with the film on the water-side, even if a substantially thicker film is found on the air-side of the
air-water interface. This means that in the case of $O_2$, $C_{air}$ is simply the saturation concentration of
$O_2$ in water, which is a well-described function of the water temperature and salinity (Garcia and
Gordon 1992) and the atmospheric pressure.

Turbulence, or turbulent-like motions, that affects or controls the thickness of the film on the
water side, and thus the diffusive resistance to gas transport, can originate from both below and
above the water. In shallow streams and rivers, this turbulence is typically generated by the water
flow over an uneven or rough bottom. Substantial heat loss from the water can similarly result in
density driven water motion that erodes the film (Bannerjee and MacIntyre 2004). On the





contrary, in reservoirs, lakes, and estuaries, the turbulence on the water side of the interface is typically generated by wind, which makes wind speed the dominant controlling variable for $k$ for

such systems (Marino and Howarth 1993). Despite the fact that typical conditions such as rough weather, surface waves, and rain can rupture the films on the water side, the simple expression for gas exchange (Eq. 1) is still applied, but with $k$ values that are adjusted accordingly (Watson et al. 1991). Keeping these multivariable, highly dynamic, and complex controls in mind, it is evident that determination of representative $k$ values for specific sites is a challenging task.


## 1.2 Formulation of problem

A number of approaches have been used to study and determine values for $k$. For smaller rivers and streams they include targeted parallel additions of volatile tracers such as propane and hydrologic tracers such as dissolved chloride, where the latter is added to correct for dilution of

propane due to hyporheic mixing (Genereux and Hemond 1992; Koopmans and Berg 2015). A common approach for smaller reservoirs and lakes relies on additions of inert tracers, e.g. $SF_6$ (Wanninkhof 1985; Cole et al. 2010), whereas floating chambers are often deployed in larger rivers, reservoirs, lakes, and estuaries (Marino and Howarth 1993). In a limited number of studies of large reservoirs and lakes, tower-mounted atmospheric eddy covariance systems have been

used to measure air-water exchange, and from that, $k$ values were derived (Anderson et al. 1999; Jonsson et al. 2008; Mammarella et al. 2015). Partly motivated by the substantial and often methodologically challenging effort required to measure $k$ at specific sites with any of these approaches, many studied have simply relied on general empirical correlations for $k$ produced by fitting measurements done in other aquatic systems (Raymond and Cole 2001; Borges et al. 2004;

Cole et al. 2010). With the exception of atmospheric eddy covariance measurements, none of these approaches represent a direct way of determining $k$ values because they rely on assumptions that often are difficult to assess, or simply not fulfilled. As a result, gas exchange is viewed among aquatic scientists as the primary source of uncertainty in many standard estimates for aquatic systems (Wanninkhof et al. 1990; Raymond and Cole 2001; Raymond et al. 2012).




### 1.3 Scope of work

The aquatic eddy covariance technique for $O_2$ flux measurements under undisturbed in situ conditions was originally developed for the benthic environment (Berg et al. 2003). The approach has several significant advantages over other flux methods, including its non-invasive nature (Lorrai et al. 2010), high temporal resolution (Rheuban & Berg 2013), and its ability to integrate over a large benthic surface (Berg et al. 2007). As a result, it has been used to measure whole-system fluxes for substrates such as river bottoms (Lorke et al. 2012; Berg et al. 2013), seagrass meadows (Hume et al. 2011; Rheuban et al. 2014), and coral reefs (Long et al. 2013; Rovelli et al. 2015).

Here, we applied the aquatic eddy covariance technique 'upside down' right below the air-water interface to measure $O_2$ fluxes. From them, we derived exchange coefficients for $O_2$, and then standard gas exchange coefficients ($k_{600}$). All measurements were done from a floating platform, and because we used a newly developed fast-responding dual $O_2$-temperature sensor (Berg et al. 2016), we get parallel fluxes of $O_2$ and thermal energy, or sensible heat. We conducted proof-of-concept tests including deployments at three river sites that were up to 40 hours long.

## 2 Methods

### 2.1 Floating measurements platform

All measurements were done from a floating platform with a catamaran-shaped hull (Fig. 1) that was kept at a fixed position at the river sites by two upstream anchors. Due to this setup and the current's constant pull on the hull, the platform was stationary during deployment. The modular design and the catamaran-shaped hull allow the platform to be collapsed for storage and easy shipment in a standard sturdy Polymer gun case (Pelican Products, USA).

The 3D velocity field was measured with an Acoustic Doppler Velocimeter (ADV) with a cabled sensor head (cabled Vector, Nortek AS, Norway). This type of ADV allowed the sensor head to be



positioned facing upwards (Fig. 1) while recording the velocity field right below the air-water

interface (typically ~5 cm). Data were collected continuously at a rate of 64 Hz and represent

water velocity values averaged over the ADV's cylindrical measuring volume (h ~1.4 cm, Ø ~1.4

cm) located 15.7 cm above the sensor head (Fig. 1).

The $O_2$ concentration was measured with a new fast-responding dual $O_2$-temperature sensor

(Berg et al. 2016) which, combined with the velocity data, allows for simultaneous fluxes of $O_2$

and sensible heat to be derived and also instantaneous temperature correction of the $O_2$ signal.

This sensor was developed specifically for eddy covariance measurements and was designed to

interface with our standard ADVs (Vectors, Nortek AS, Norway) through a single cable supplying

power to the sensor and also transmitting its two outputs, one for $O_2$ and one for temperature, to

the ADV's data logger to be recorded along with velocities to ensure perfect time alignment of all

data. The $O_2$ measuring part of this new sensor is a small planar optode and concentrations are

determined from fluorescence life-time measurements (Klimant et al. 1995; Holst et al. 1997;

Holst et al. 1998). The sensor tip size, including the temperature thermistor, has a diameter of 8.0

mm which makes it far more robust than $O_2$ micro-sensors typically used for aquatic eddy

covariance measurements and its response times ($t_{90\%}$) were measured to be 0.51 s and 0.34 s for

$O_2$ and temperature, respectively (Berg et al. 2016). The edge of the sensor tip was positioned

~2.0 cm downstream of the edge of the ADV's measuring volume so that water passed through

this volume before sweeping over the angled $O_2$ sensing tip (Fig. 2a). This setup ensured

undisturbed measurements of the natural current flow. Power was supplied from an external

battery (Fig. 1a) with a capacity that allowed 64 Hz data to be collected continuously for at least

48 h. Because all instrument components were designed for underwater use they are not affected

by rain or humid conditions.

Measurement of supporting environmental variables during each deployment allowed

verification of recorded data and assisted in the interpretation of the derived eddy fluxes. These

variables included mean $O_2$ concentration and temperature at the measuring depth recorded



every 1 min with one or two stable independent dual O$_2$-temperature sensors (miniDOT, PME, USA). In some deployments photosynthetically active radiation (PAR) was also recorded at the measuring depth every 5 min using an independent submersible PAR sensors (Odyssey, Dataflow

Systems, New Zealand). For one deployment, light data were taken from nearby meteorological weather stations.

## 2.2 Field tests

The new approach for determining air-water gas exchange rates and associated exchange

coefficients from underwater eddy covariance measurements was tested at three river sites, one in the Hardware River (Virginia) and two in the Mechums River (Virginia). All sites had a fairly linear run with a water depth between ~0.3 and ~1 m and smooth and quietly flowing water (Fig. 1c) without standing riffles or waves. Typical surface flow velocities ranged from 6 to 30 cm s$^{-1}$. The ADV and the fast-responding O$_2$-temperature sensor were adjusted to record data ~5 cm

below the air-water interface. Four deployments lasting up to 40 h were initiated on November 22, 2015 and September 14, 2016 in the Hardware River and on December 21, 2016 and January 18, 2017 in the Mechums River. Using a level and by placing dive weights on the platform (Fig. 1b) care was taken to ensure that the instrument was as level as possible to minimize post-processing rotations of the velocity field to correct for sensor tilt.


## 2.3 Calculations of eddy fluxes

Fluxes of O$_2$ and heat were extracted from the raw eddy covariance data following the same multi-step process briefly described below for O$_2$. If needed, the O$_2$ concentration was calibrated against the stable independent dual O$_2$-temperature sensor data. All 64 Hz data were then reduced to 8

Hz data, which reduces noise while providing sufficient resolution to contain the full frequency spectrum carrying the detectable flux signal (Berg et al. 2009). This assumption was validated by comparing fluxes calculated from both 8 and 64 Hz data for a subset of the data.





O₂ fluxes, one for each 15-min data segment, were extracted from the 8 Hz data using the software

package EddyFlux version 3.1 (P. Berg unpubl.). If needed, this software rotates the flow velocity

field for each data segment to correct for any sensor tilt (Lee et al. 2004; Lorrai et al. 2010; Lorke

et al. 2013) bringing the transverse and vertical mean velocities to zero. The vertical eddy flux

was then calculated as (defined positive upward):


$$J_{eddy} = \overline{w'C'} \tag{2}$$

where the overbar symbolizes the averaging over the 15-min data segment, and $w'$ and $C'$ are the

fluctuating vertical velocity and the fluctuating O₂ concentration, respectively. These fluctuating

components are calculated as $w - \bar{w}$ and $C - \bar{C}$ where $w$ and $C$ are measured values (at 8 Hz), and

$\bar{w}$ and $\bar{C}$ are mean values defined as least square linear fits to all $w$ and $C$ values within the 15-min

time segment, a procedure usually referred to as linear de-trending (Lee et al. 2004; Berg et al.

2009).

Due to the response time of the dual O₂-temperature sensor and its position downstream from the

ADV's measuring volume, a time shift correction was applied. This was done by repeating the

outlined flux extraction procedure, while shifting the 8 Hz O₂ concentration data back in time, 1/8

s at a time, until the numerically largest flux was found.

Estimating the gas exchange coefficient requires the O₂ flux over the air-water interface to be

known. However, the eddy flux, $J_{eddy}$, is measured ~5 cm below the interface. By using the linear fit

to the measured O₂ concentrations in each 15-min data segment, defined as $\bar{C}$ above, $J_{eddy}$ is

corrected for storage of O₂ in ~5 cm volume of water to give the flux at the interface:

$$J_{eddy, air-water} = J_{eddy} - \int_0^h \frac{d\bar{C}}{dt} dz \tag{3}$$






where $h$ is the ~5 cm tall water column. For further details on this flux extraction protocol included in EddyFlux version 3.1, see Lorrai et al. (2010), Hume et al. (2011), and Rheuban et al. (2014). For presentation, the 15-min fluxes were lumped in groups of four to give hourly values.

To examine the eddy frequencies that carried the flux signal, for several consecutive data segments in each deployment, cumulative co-spectra of the $O_2$ concentration and the vertical velocity were calculated using the software package Spectra version 1.2 (P. Berg unpubl.).

**2.4 Calculations of gas exchange coefficients**

The saturation concentration of $O_2$ ($C_{air}$ in Eq. 1) was calculated from Garcia and Gordon (1992) as a function of salinity (here 0 ppt) and surface water temperature measured with the fast-responding dual $O_2$-temperature sensor ~5 cm below the air-water interface and then corrected for atmospheric pressure using Henry's law (average sea-level pressure of 1013.25 mbar corrected for elevation). The water column $O_2$ bulk concentration ($C_{water}$ in Eq. 1) was measured

with the same sensor. By substituting $J_{air-water}$ (Eq. 1) with the 15-min values for $J_{eddy,\ air-water}$ (Eq. 3), a gas exchange rate for $O_2$ was calculated from Eq. 1 and converted to the standard exchange coefficient, $k_{600}$, for $CO_2$ at 20 °C (Jähne et al. 1987; Wanninkhof 1992; Cole et al. 2010). For presentation, the 15-min $k_{600}$ values were lumped in groups of four to give hourly values.

**3 Results**

All four deployments resulted in high-quality time series of the velocity field, the $O_2$ concentration, and the temperature ~5 cm below the air-water interface, and derived from those, air-water fluxes of $O_2$ and heat, and gas exchange coefficients. These data and their interpretation are presented below.


**3.1 Data example**

For a 40 h long deployment initiated on January 18, 2017 in the Mechums River, the 15-min mean current velocity (Fig. 2a) was relatively constant, averaging 20.5 cm s$^{-1}$. The $O_2$ concentration



measured with the fast-responding dual $O_2$-temperature sensor (Fig. 2b) agreed closely with the

concentration recorded with the independent sensor and showed a distinct diurnal pattern.

During most of the first night of the deployment, the $O_2$ concentration increased linearly (h 19 to h

32), whereas a smaller and non-linear increase that tapered off was measured during the second

night (h 45 to h 56). A diurnal pattern was also seen in the calculated $O_2$ saturation concentration

(Fig. 2b) reflecting variation in water temperature. The cumulative $O_2$ flux (Fig. 2c), with each

data segment covering a 15-min time interval, had clear linear trends indicating a strong eddy flux

signal in the data. The hourly $O_2$ flux (Fig. 2d), representing means of four successive 15-min flux

estimates, also exhibited a clear diurnal pattern with a nighttime average uptake by the river of

16.4 mmol $m^{-2}$ $d^{-1}$ for the first night, 9.1 mmol $m^{-2}$ $d^{-1}$ for the second night, and an average daytime

release of 11.1 mmol $m^{-2}$ $d^{-1}$. As observed for the $O_2$ concentration (Fig. 2b), the hourly $O_2$ flux

differed during the two nighttime periods with a close-to constant flux during the first night and a

flux that tapered off during the second night. The hourly standard gas exchange coefficient ($k_{600}$,

Fig. 2e) derived from the hourly $O_2$ flux (Fig. 2d) and the $O_2$ concentration difference over the air-

water interface (Fig. 2b) was almost constant over the first night of the deployment with an

average of 3.9 m $d^{-1}$. After that, $k_{600}$ diminished almost 3-fold to a value of 1.4 m $d^{-1}$ during the

daytime. During the second night, $k_{600}$ tapered off markedly from a level found for the first night

to almost 0.89 m $d^{-1}$ during the last four h of the deployment. This pattern was unexpected given

the almost constant mean current velocity and insignificant winds (Fig. 2a) and the similar $O_2$

concentration difference (Fig. 2b) for the two nighttime periods. The pattern suggests that gas

exchange was controlled by a more dominant driver than the river current velocity or winds (see

Discussion below).

The parallel results derived from the temperature data measured with the fast-responding dual

$O_2$-temperature sensor agreed perfectly with the temperature recorded with the stable

independent sensor (Fig. 3b) and had, as with the $O_2$ concentration, a distinct diurnal pattern. A

close-to linear decrease occurred during the first night (h 18 to h 32) whereas a smaller and non-

linear decrease that tapered off was recorded during the second night (h 45 to h 56). During the



daytime the temperature increased. Unfortunately, we do not have reliable on-site measurements of the air temperature, but we infer that it, together with shortwave (sunlight during day) and longwave (nighttime) thermal radiation, controlled the recorded water temperature variations

(Fig. 3b). The cumulative heat flux (Fig. 3c) had, as for $O_2$, clear linear trends indicating a strong flux signal in the data. The hourly heat flux (Fig. 3d) also exhibited a clear diurnal pattern with a nighttime average release of heat of 60.6 W m$^{-2}$ for the first night and 27.5 W m$^{-2}$ for the second night. As was observed for the temperature (Fig. 3b), the hourly heat flux showed different trends for the two nights with a close-to constant flux during the first night and a flux that tapered off

during the second night.

Ignoring differences in the sign, representative cumulative co-spectra for the $O_2$ and heat flux (Fig. 4) during the first night (Figs. 2, 3) were similar in the 0.1 to 1 Hz frequency band with all substantial flux contributions for both the $O_2$ and heat flux having frequencies lower than ~0.9 Hz.

This result, combined with the fast-responding dual $O_2$-temperature sensor's response times ($t_{90\%}$) of 0.51 s for $O_2$ and 0.34 s for temperature (Berg et al. 2016), indicates that the entire eddy flux signal over all frequencies was accounted for in our measurements.

**3.2 Representative gas exchange coefficients**

The three other test deployments were shorter than the one presented in Figs. 2 and 3 but results were of comparable quality. Average values for selected parameters covering periods of time with several successive 15-min time intervals from all four deployments are given in Table 1. These periods were identified by containing consecutive time intervals with consistent standard gas exchange coefficient values, $k_{600}$, that had little variation and appeared to represent a particular

field condition. The longest period ($n = 51$) covers the first full night of the deployment shown in Fig. 2 (h 19 to h 32). Overall, the average current velocity varied from 8.3 to 28.4 cm s$^{-1}$ while $k_{600}$ ranged from 0.4 to 5.1 m d$^{-1}$, or more than a factor of 12.





There was no significant relationship (r = 037, p = 0.22) between river current velocity and $k_{600}$
values (Fig. 5) for all the data in Table 1. Substantial variations in $k_{600}$ values were found for some
individual deployments even though the current velocity did not change markedly. Most
prominently in the Mechums River deployment, site b (Figs. 2, 3), where the $k_{600}$ values varied
more than a factor of 5. As we discuss below, this suggest that, at least for some sites and under
some field conditions, other drivers of air-water gas exchange than river flow and winds are more
important.

### 3.3 Temperature effects on $O_2$ readings – a possible methodological bias

Highly sensitive fast-responding $O_2$ sensors that can be used for aquatic eddy covariance
measurements are inherently sensitive to temperature variations, and thus they will give variable
readings at the same molar $O_2$ concentration if the temperature changes. Typical temperature
coefficients (% change in $O_2$ concentration reading caused by a temperature change of 1 °C) have
values of ~3 % (Gundersen et al. 1998). This implies that rapid temperature fluctuations
associated with a turbulent heat flux will mistakenly be recorded as fluctuations in $O_2$
concentration and bias the eddy flux calculation unless a temperature correction of the $O_2$ reading
is performed. In this study, we relied on a new fast-responding dual $O_2$-temperature sensor (Berg
et al. 2016) which puts out rapid simultaneous readings of both the $O_2$ concentration and the
temperature within a distance of few mm and makes this correction possible. This was done for
all $O_2$ fluxes we present. Below, we exemplify the nature and magnitude of this bias if this
correction is omitted using data measured during the first night of the deployment depicted in
Figs. 2 and 3.

The turbulent temperature fluctuations for a 3-min period shown in Fig. 6a are associated with a
vertical heat flux of ~60 W m$^{-2}$ (Fig. 3d) and amount to ± ~0.015 °C. Based on a temperature
coefficient of ~3 %, according to the $O_2$ calibration equation provided by the producer of our
sensor (JFE Advantech, Japan), this translates into fluctuations in $O_2$ concentration readings of ±
~0.2 µmol L$^{-1}$ (Fig. 6a). Using such 'simulated' $O_2$ data, representing solely temperature sensitivity



effects and produced from the 8 Hz nighttime temperature data (h 18 to h 32, Fig. 3), to calculate

an $O_2$ flux bias gives a release of 11.9 mmol m$^{-2}$ d$^{-1}$ (blue bar, Fig. 6b). Using the temperature

corrected $O_2$ data, as was done for all other calculations we present, gives an oppositely directed

$O_2$ uptake of 16.9 mmol m$^{-2}$ d$^{-1}$ (red bar, Fig. 6b), whereas using the $O_2$ readings, but without the

rapid temperature correction, gives a release of only 4.4 mmol m$^{-2}$ d$^{-1}$ (green bar, Fig. 6b).

## 4   Discussion

Deploying the aquatic eddy covariance technique right below the air-water interface provided a

feasible way to determine gas exchange rates and coefficients. Relative to what is possible with

traditional methods, this new approach gives gas exchange rates and coefficients with an

improved precision and at a higher spatial and temporal resolution. For those reasons, the

approach has the potential to enhance our knowledge on the dynamics and controls of gas

exchange and thus benefit aquatic ecosystem studies and pave the way for new lines of ecosystem

research.

These points are exemplified in our longest test deployment that lasted 40 h (Figs. 2, 3) and

resulted in aquatic eddy covariance data for both $O_2$ and temperature of a quality and internal

consistency that fully match those published for many benthic environments (see review by Berg

et al. 2017). Specifically, the 8 Hz velocity, $O_2$, and temperature data (Figs. 2a, 2b, 3b) were

recorded with low noise and perfectly matched measurements with the stable independent

sensor (Figs. 2b, 3b). Furthermore, the cumulative fluxes (Figs. 2c, 3c) had clear linear trends that

indicate a strong and consistent flux signal in the data, and the times where the hourly $O_2$ flux

changed direction (Fig. 2d, positive values represent a release), matched exactly the times where

the driving $O_2$ concertation difference changed sign (Fig. 2b). Moreover, the cumulative co-spectra

for the $O_2$ and the temperature fluxes (Fig. 4) have the shape typically seen for shallow-water

environments (Lorrai et al. 2010; Berg et al. 2013). Finally, for both $O_2$ and temperature there was

a clear relationship between the flux over the air-water interface (Figs. 2d, 3d) and the observed

change in the water column (Figs. 2b, 3b). For $O_2$, for example, the ratio between the averaged





fluxes for the two nights (h 21 to h 30 vs. h 45 to h 54, Fig. 2d) equals 2.0 which is close to the

ratio of 2.2 between the changes in water column concentrations (Fig. 2b) for the same two

periods.

Both the $O_2$ and the temperature data (Figs. 2b, 2d, 3b, 3d) contained a clear diurnal signal overall.

For $O_2$, however, this was not driven by biological processes, i.e. net primary $O_2$ production during

daytime and respiration during nighttime, as this would have resulted in an increase in mean

water column $O_2$ concentration during daytime and a decrease at nighttime. That the opposite

pattern was found indicates that physical processes related to thermal conditions were

controlling $O_2$ dynamics. Specifically, colder nighttime air temperatures and possibly also long-

wave thermal radiation to the atmosphere were driving the substantial heat flux out of the river

(Fig. 3d) which resulted in the falling water temperatures (Fig. 3b). This, in turn, was changing the

$O_2$ saturation concentration ($C_{air}$ in Eq. 1) and thus the driving concentration difference of $O_2$

exchange over the air-water interface (Fig. 2b). During the daytime, the reverse pattern was in

place. This rather complex relationship, or linkage via physical processes, is the only mechanism

that can explain the overall pattern found for this deployment (Figs. 2, 3). Considering that these

measurements were done under conditions that did not include any uncommon or extreme

weather conditions suggests that physical processes, and not biological processes, are often an

important, or even the main, driver of $O_2$ dynamics in shallow-water rivers and streams. An

unfortunate consequence of this dominance or control by physical conditions, which we believe is

not yet fully recognized, is that it adds substantial uncertainty to the widely used approach of

deriving metabolic estimates (e.g., gross primary production, respiration, net ecosystem

metabolism) from time series of measured water column $O_2$ concentrations (Odum 1956; Hall et

al. 2016).

The standard gas exchange coefficients ($k_{600}$) for all of our four deployments (distributed on three

different river sites, all with smooth quietly flowing water without standing riffles or waves, Fig.1)

did not show a significant relationship with river current velocity (Fig. 5, Table 1). This is in line



with previously published results from across-site comparisons (Hall et al. 2016), but the

substantial variation among $k_{600}$ values for some individual deployments (in particular for the

'Mechums River, site b' deployment, Figs. 2d) despite only moderately varying river flow velocity

and insignificant winds is surprising. For example, $k_{600}$ varied from a close-to constant value of 3.9

m d$^{-1}$ during the first night (h 19 to h 32, Fig. 2e), followed by an almost 3 times smaller daytime

value of 1.4 m d$^{-1}$ (h 33 to h 42), and then increased again at the onset of the second night before

finally tapering off to a small value of 0.9 m d$^{-1}$ (h 52 to h 56) at the end of the deployment. The co-

variance of the heat exchange (Fig. 3d) suggests that turbulence, or turbulent-like motions (which

stimulates gas exchange) was generated by natural convective forces driven by the substantial

heat-loss from the river during the nighttime (Fig. 3d). Conversely, during the daytime, when the

heat flux was directed into the river (Fig. 3d), turbulent motions were presumably dampened by

vertical temperature stratifications in the surface water. Given the 'low-energy' smooth and

quietly flowing water, we find this explanation for the varying $k_{600}$ values (Fig. 2e) likely and note

that it has been described before (Bannerjee and MacIntyre 2004; MacIntyre et al. 2010). We also

note that this observed complex pattern illustrates the difficulties that can be associated with

determining accurate air-water gas exchange rates and coefficients without direct site- and time-

specific measurements.


An important methodological finding linked to the new approach is that $O_2$ sensor readings

should, at least in some cases, be corrected using concurrent rapid temperature readings as was

done here for all $O_2$ fluxes used to estimate air-water gas exchange coefficients (Fig. 2, Table 1). In

the benthic environment where the aquatic eddy covariance technique for $O_2$ flux measurements

was firsts developed (Berg et al. 2003), the vertical turbulent heat flux is usually small relative to

the $O_2$ flux due to slowly and modestly varying mean temperatures in the bottom water. At the

air-water interface, however, the heat flux is typically larger due to substantial variations in air

temperatures and short- and long-wave thermal radiation. As a result, rapid temperature

fluctuations associated with the turbulent heat flux below the air-water interface will mistakenly

be recorded as fluctuations in the $O_2$ concentration if this correction is omitted and bias the $O_2$



flux calculation substantially (Fig. 6). It is unclear how widespread this problem is – more studies are needed to determine that – but in the example included here, this bias alters the flux by more than a factor of 2 (Fig. 6). Our data were recorded during winter, and one could argue that the $O_2$ exchange would be much larger during summer due to extensive primary production and

respiration which would reduce the relative magnitude of this bias (Fig. 6). But as the $O_2$ flux is indeed likely to be more pronounced during summer than during winter, so is the heat flux.

## 5   Summary and recommendations

Based on our proof-of-concept deployments, the aquatic eddy covariance technique applied right

below the air-water interface should be particularly useful in detailed studies of gas exchange that evaluate its dynamics and controls. The approach can consequently help reduce the generally recognized problem of large uncertainties linked to gas exchange estimates in traditional aquatic ecosystem studies.

The floating platform we used here for measuring aquatic eddy covariance fluxes right below the air-water interface (Fig. 1) can easily be reproduced as it relies exclusively on standard materials and commercially available instrumentation, the latter designed with plug-and-play capabilities. Furthermore, standard software for eddy flux extractions developed for the benthic environment or for the atmospheric boundary layer can be used to estimate air-water fluxes.


We recommend that eddy covariance data are recorded close to the air-water interface (Fig. 1c) to minimize the effects of the $O_2$ storage in the water between the measuring point and the surface and because gradients of both $O_2$ and temperature can form in the upper water column. We also recommend that simultaneous rapid temperature measurements are performed within a few mm

of the $O_2$ concentration recordings to allow for temperature corrections of the $O_2$ signal (Fig. 6).

Finally, our results illustrate that the $O_2$ concentration difference driving the air-water gas exchange is often small (Fig. 2), here < 2 % of the absolute concentration. This emphasizes the




importance of relying both on accurately calibrated sensors to measure the water bulk

concentration ($C_{water}$ in Eq. 1) and precise determinations of the saturation concentration ($C_{air}$ in Eq. 1) that is corrected for temperature, salinity, and atmospheric pressure.

## 6  Future work

A further development of the new application of the aquatic eddy covariance technique presented

here is to perform similar measurements from a moving platform in small lakes, reservoirs, and estuaries. In these environments, gas exchange and gas exchange coefficients are expected to vary spatially, for example from the lee to windward side of the aquatic system. By using a floating autonomously moving platform, we anticipate that such variations can be spatially mapped out and studied. We are currently performing the first tests along these lines.


## 7  Acknowledgements

This study was supported by grants from the National Science Foundation (ESC-1550822, OCE-1334848) and the University of Virginia. We thank Julie and John Baird, Nancy and Ed Mcmurdo, Martha Hodgkins, and Brian Richter who allowed us to work on their beautiful properties in the

Hardware River and the Mechums River. Finally, we thank Rachel E. Michaels for editorial assistance on the manuscript.



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





## 9 Tables

| Deployment | Start date | *n* | Current velocity | O$_2$ flux | $k_{600}$ |
| --- | --- | --- | --- | --- | --- |
| - | - | - | cm s$^{-1}$ | mmol m$^{-2}$ d$^{-1}$ | m d$^{-1}$ |
| Hardware River, dep. 1 | Nov 22, 2015 | 20 | 28.4 | 9.1 | 1.6 |
| " | " | 39 | 27.5 | 12.0 | 2.7 |
| " | " | 13 | 27.6 | -10.7 | 2.5 |
| Hardware River, dep. 2 | Sep 14, 2016 | 20 | 8.7 | 7.0 | 0.4 |
| " | " | 4 | 8.3 | 9.4 | 0.7 |
| Mechums River, site a | Dec 21, 2016 | 23 | 9.4 | -42.9 | 2.3 |
| " | " | 36 | 9.3 | -29.2 | 1.7 |
| Mechums River, site b | Jan 18, 2017 | 4 | 25.6 | -8.9 | 1.9 |
| " | " | 51 | 18.4 | 16.8 | 3.9 |
| " | " | 34 | 20.4 | -11.8 | 1.3 |
| " | " | 3 | 22.9 | 19.3 | 5.1 |
| " | " | 16 | 23.4 | 10.1 | 2.1 |
| " | " | 26 | 21.3 | 5.8 | 1.0 |

**Table 1:** Representative standard gas exchange coefficients ($k_{600}$) along with current velocity and O$_2$ flux for four deployments at three different sites. The third column (*n*) specifies the number of 15-min time intervals included in the averages. Values from the last deployment (Mechums River, site b) are depicted in Figs. 2 and 3.





## 10 Figures

**Figure 1:** Floating platform for determining air-water gas exchange. **(a)** The 120 cm long and 90 cm wide platform with a catamaran-shaped hull being prepared for deployment. Four inflatable fenders provide flotation. **(b)** The platform deployed in the Hardware River and anchored to both river banks. A dive weight is used to level the platform. **(c)** Close-up look at the ADV's three-pronged upward-facing sensor head and the fast-responding dual $O_2$-temperature sensor. Two stable independent dual $O_2$-temperature sensors used for calibration are seen to the far right.





**Figure 2:** Forty h long test deployment initiated at 16:00 in the afternoon as indicated on the x-axis. **(a)** Three velocity components at 8 Hz (*x*, *y*, *z*; *z* is vertical) and 15-min mean current velocity. **(b)** $O_2$ concentration at 8 Hz measured with the dual $O_2$-temperature sensor and at 1-min measured with an independent sensor. **(c)** Cumulative flux over 15-min time intervals with clear linear trends. **(d)** Hourly $O_2$ flux (positive values represent a release from the river), each value based on 15-min flux extractions (n = 4, SE) and day light measured at a nearby weather station. **(e)** Hourly standard gas exchange coefficient ($k_{600}$) based on 15-min estimates (n = 4, SE). The few gaps in the data are for the times where the driving $O_2$ concentration difference changes sign (panel **c**).





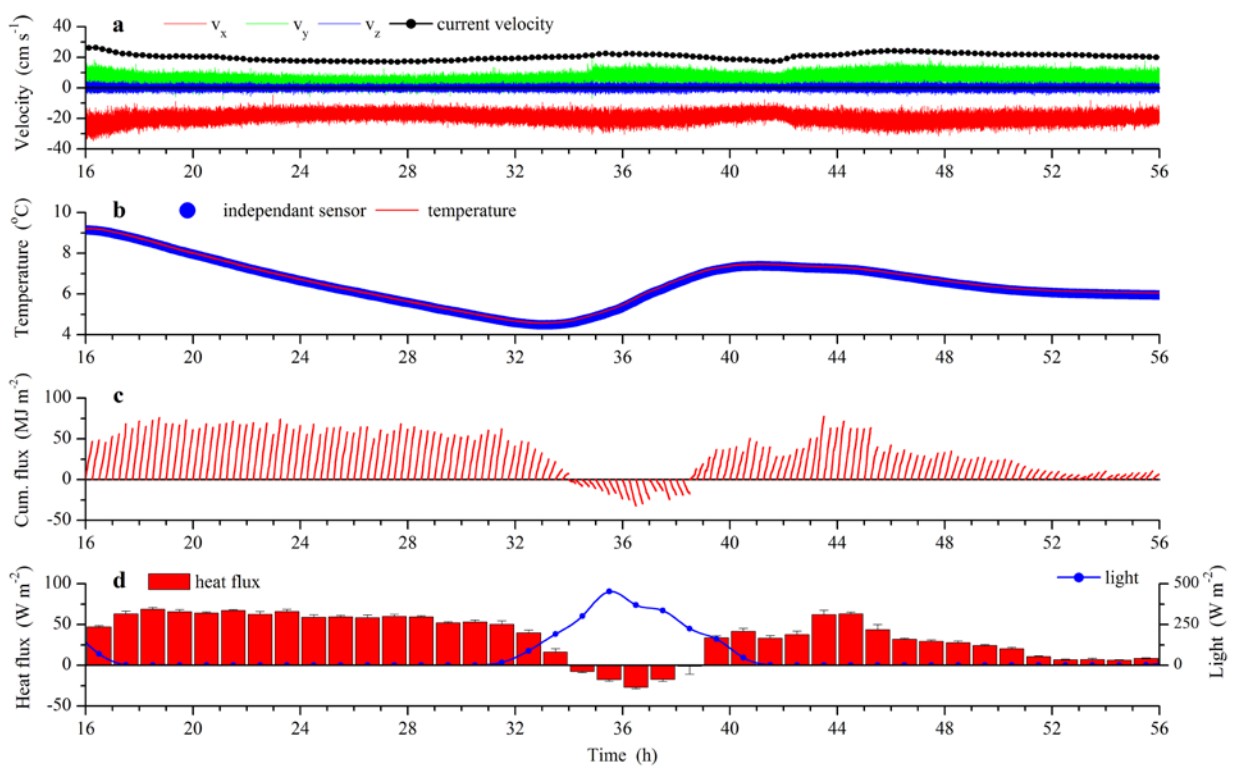

**Figure 3:** Same deployment as in Fig. 2, but with results for temperature and heat. The deployment was initiated at 16:00 in the afternoon as indicated on the x-axis. **(a)** Three velocity components at 8 Hz ($x$, $y$, $z$; $z$ is vertical) and 15-min mean current velocity. **(b)** Temperature at 8 Hz measured with the dual $O_2$-temperature sensor and at 1-min measured with an independent optode. **(c)** Cumulative flux over 15-min time intervals with clear linear trends. **(d)** Hourly heat flux, each value based on 15-min flux extractions (n = 4, SE) and day light measured at a nearby weather station. Positive flux values represent a release of heat from the river.





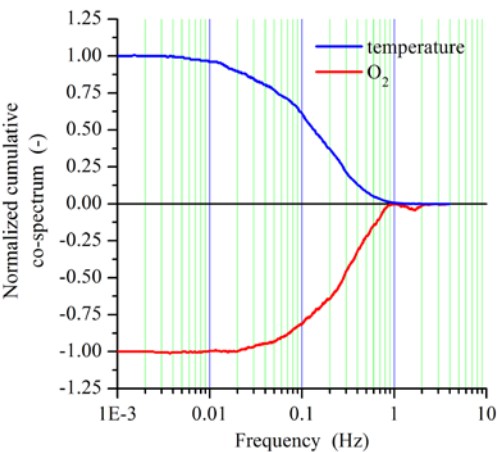

**Figure 4:** Nighttime normalized cumulative co-spectra for the vertical velocity combined with the $O_2$ concentration and the temperature, respectively, revealing which frequencies carried the eddy flux signal.



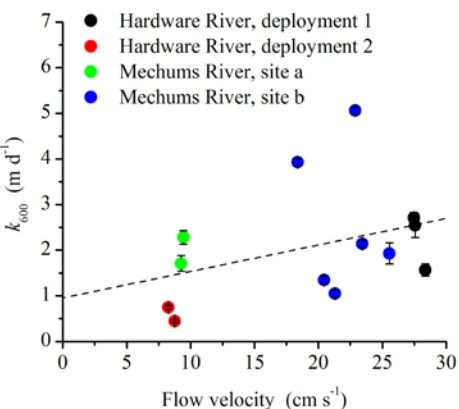

**Figure 5:** Standard gas exchange coefficient, $k_{600}$, plotted against river current velocity. The dotted line is a linear fit to all data ($R = 0.37$, $p = 0.22$).





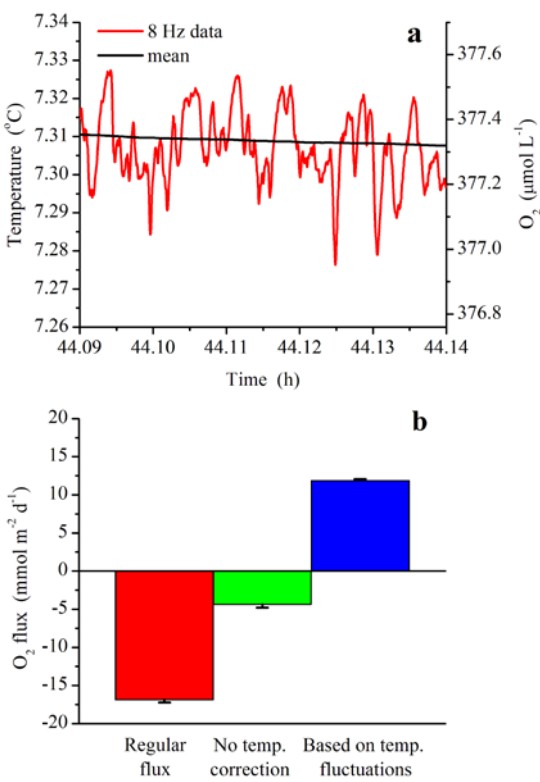

**Figure 6:** Bias that can arise if $O_2$ concentration sensor readings are not corrected using rapid parallel temperature measurements. **(a)** Recorded 8 Hz data of temperature fluctuations and their mean (left axis) through 3-min and the resulting fluctuations in $O_2$ concentration that would be recorded by a sensor with a temperature coefficient of 3 % (right axis). **(b)** Average air-water fluxes, all for the first night (h 18 to h 32) of the deployment depicted in Figs. 2 and 3, calculated using temperature corrected data (red bar), data without rapid temperature correction (green bar), and 'simulated' data produced from rapid temperature recordings as shown in panel **a** and assuming a temperature coefficient of 3 % (blue bar).