# Peer review of "Continuous measurement of air-water gas exchange by underwater eddy covariance"

_Biogeosciences, 2017_

## Referee Comment (RC1) · Anonymous Referee #1 · 13 Sep 2017

This manuscript describes an important methodological advance for aquatic sciences by demonstrating that the eddy covariance method can be applied from the water-side of the air-water interface to measure oxygen and heat fluxes and to derive standard gas exchange coefficients. The method is used successfully at three shallow river sites where physical processes, especially heat exchange, are found to drive diurnal variations in gas exchange. The paper is well organized with careful, highly reasoned arguments for the approach and data treatments. The data examples are clear and mostly convincing.

The only troubling part of the paper is sections that describe the possible methodolog-

ical bias produced by temperature effects on the O2 sensor time series and how the authors have corrected their measurements for this bias. Although I agree this bias is likely and needs to be understood, I do not think the authors have shown they really know its magnitude. They estimate a ∼3% change in the oxygen reading per 1 degree C, and even with relatively small temperature fluctuations (<0.1 oC) this creates a bias about 3 times the measured signal (Figure 6). What if the effect was 4% or 2% instead of 3%? How consistent is the effect between optode sensors and their films? Is the effect proportional to the oxygen concentration or independent of oxygen concentration? Since the response time of the thermistor is faster than the optode, does this alter the correction? In short, the authors need to independently measure the magnitude of the temperature effect before applying a correction. This might be done with experiments where the oxygen partial pressure is held constant but temperature varied. Otherwise, the applied corrections may be creating more bias than they are removing.

If the authors can address the above concern any remaining revisions to the paper will be minor. Below are listed areas by line number that might be clarified.

Line 25. I question whether it is known that physical controls are "prevalent in lotic systems". Perhaps it would be better to say "can be prevalent in lotic systems and adds uncertainty to assessments of biological activity for such systems that are based on water column O2 concentration changes".

Line 30. What is meant by "erosion in the surface water"? Erosion of what?

Lines 78-80. Awkward sentence. Please restructure.

Line 87. Omit "but" in this sentence.

Line 93. Indicate where and how the tracer additions are made.

Lines 103-104. Change "studied" to "studies" and then clarify what is meant by "fitting measurements done in other aquatic systems".

Line 108. "many standard estimates" of what? Please clarify. Are you talking about

carbon budgets?

Line 125. Reword as: "we were able to derive parallel fluxes. . ."

Line 126. Reword as: "proof-of-concept tests that were up to 40 hours long at three river sites."

Line 131. Reword as: "All measurements were made from. . ."

Line 140. How was the measurement position ∼5 cm below the interface determined? Later (line 222) can you indicate how sensitive the storage term correction is to changes in this measurement?

Line 144. Why is the senor not identified as manufactured by Rinko?

Line 155. Separate into two sentences here. Indicate how reproducible the response times are with each fresh sensor film put on the optode tip.

Line 158. Why reference Fig. 2a here?

Line 169. Change to "PAR sensor".

Line 183. "as level as possible" is vague. Can you indicate within a certain number of degrees from vertical? Please clarify how tilt changes were corrected for within the time span of a 15-min burst as the sensor must bob up and down some.

Line 191. The key word here is "detectable". There may still be high frequency signals lost because they are not detectable by these sensors.

Line 227. It would be helpful here if the authors gave more information about how the "Spectra version 1.2" code treats the data. Also, what is meant by "several consecutive data segments"? How does this relate to what is shown in Fig. 4?

Line 237. It would be helpful for the authors to present the relationships for calculating k600 even though they are in the papers cited.

Line 256. Since the data is presented as hourly fluxes, why not change the units in the

figures to per hour rather than per day?

Line 269. Suggest reword as: "controlled by a driver apart from the river current velocity or winds. . .".

Line 272. Unclear what results are being referred to here.

Line 317. I do not see why the authors reference Gundersen et al. 1998 here? This paper discusses the temperature sensitivity of oxygen microelectrodes that operate by different principles than optodes. The microelectrode temperature effect is usually related to the gas solubility in the membrane and changes in the diffusion rate.

Lines 352-355. Good argument here. Correct the spelling of "concentration" in line 355.

Line 371. Reword as: "This, in turn, changed the. . .."

Line 380-383. Can the authors take this argument further perhaps with an illustrative example?

Line 399. Do the authors have any temperature profiles from their sites that may illustrate temperature stratification during the day?

Line 410. Change to: "was first developed"

Lines 416-417. It is unsatisfying that the authors call for more studies of the temperature bias. As noted above, they need to include more concrete studies in the context of this paper.

Line 436-437. It would be helpful if earlier in the paper the authors indicated the magnitude of the O2 storage term relative to Jeddy (Equation 3).

Table 1. Add standard deviations to the parameters in the right three columns.

Figure 1c. Add arrows to indicate each identified item and indicate that the "independent dual O2-temperature sensors" are the miniDOT sensors and the sensor used for

EC is a Rinko sensor.

---

## Referee Comment (RC2) · Anonymous Referee #2 · 23 Sep 2017

The manuscript by Peter Berg and Michael Pace investigates air-water gas exchange at three shallow river sites. The authors focus on the determination of oxygen and temperature exchange by using the eddy covariance technique on a floating platform to assess gas exchange coefficients. The major findings show that oxygen dynamics (on an hourly scale) are largely independent of current velocities and biological activity; instead oxygen dynamics are driven by heat exchange, i.e. changing oxygen saturations. Furthermore, the authors point out the importance of high resolution temperature measurements to correct for the oxygen sensor specific temperature sensitivity.

The manuscript is well written / structured and of interest for a broader readership as the results have important implications for the growing community that uses the aquatic eddy covariance technique. The approach to determine gas exchange coefficients using the temperature+oxygen eddy covariance technique is also a methodological advance. However, the manuscript is lacking some important details and it would benefit from a more in-depth analysis of the interesting and promising dataset. The heat-exchange driven oxygen fluxes are only masking the more interesting biogeochemical processes and are overemphasized. In the current version, the discussion about the temperature bias raises more questions than it actually resolves. See below for a detailed argumentation.

Argumentation

1. One of the key findings and also a major part of the discussion is that heat exchange is driving most of the oxygen dynamics in shallow-water rivers. This is reasonable on timescales of hours, however, the physical process is only masking the biogeochemical processes which are still occurring and which are of importance. Based on the dataset it should be easily possible to distinguish between the biologically induced oxygen fluxes and the heat exchange induced oxygen fluxes. On the long run, the heat exchange induced fluxes should also average out implying a limited role for net exchange fluxes. When the authors follow my recommendation they could subtract the heat exchange induced oxygen flux from the total flux. I am convinced that this procedure will reveal good correlations with parameters like flow velocity and biological activity.

2. The effect of temperature fluctuations on the oxygen measurement is convincing but in the current version of the manuscript it raises several question that need to be addressed:
Resolution: The authors are discussing the response time of the temperature sensor, which is indeed in the range of the oxygen sensor. However, the sensor tip is much thicker (8mm, line 154) which implies that the spatial resolution is limiting the minimum

eddy size, i.e. frequency, that can be resolved.

Sampling Rate / Correction Procedure: The "real" sampling rate of the oxygen sensor and temperature sensor differ as the response times are slightly different and there is also a distance between the two sensors. How did the authors ensure that the temperature measured is similar to the one at the oxygen sensor tip? Did the authors also apply a time shift correction?

Range of Error: The example depicted in Figure 6 indicates that in the case of systems with large heat exchange, basically all measurements without temperature-correction are wrong. Therefore, this kind of correction needs a careful assessment. It would be interesting to see the temperature correction applied in Figure 2 for the hourly oxygen fluxes.

Figure / table / line specific comments

Figure 5: It is of interest to present the missing correlation between the gas exchange coefficient and flow velocity, however this should be contrasted by an existing correlation. An example could be the comparison of the temperature gradient versus the gas exchange coefficient. This correlation would strengthen the argumentation.

Table 1: Most of the oxygen flux is driven by heat exchange, which shows most of its variation on a daily basis. The presented oxygen fluxes are averaged in time intervals of 1 hour – 12 hours and are, therefore, strongly biased. As a result the variability within the oxygen fluxes is arbitrary as it only depends on the cut-off time.

Line 103: many studies

Line 135: It is not very convincing that a floating platform is stable when fixed as described. Actually, I would expect movements that are in the range of the eddies that carry the oxygen signal.

Line 185 and 196: How accurate was the positioning / how big was the sensor tilt? It is not clear if the correction for the sensor tilt was performed or not.

Line 210: It should be stated in which range the time shift is. Considering the very constant flow velocity and the known response times it should be possible to calculate

it. The time shift should not be bigger than the time it needs to travel from the ADV measuring volume to the sensor tip + response time!?

Line 219: Equation 3 is not adequately described, what does the second term imply, how is it measured, what is the range relative to the eddy covariance flux.

Line 239: To my knowledge "lumped" is not a statistical method.

---

## Author Comment (AC1) · 2 Nov 2017

**Response to Anonymous Referee #1:**

Based on comments from Referee #1 we plan to revise our manuscript as outlined below. All comments from the Referee are written with 'Italic' font. Our responses are written with 'normal' font.

*This manuscript describes an important methodological advance for aquatic sciences by demonstrating that the eddy covariance method can be applied from the water-side of the air-water interface to measure oxygen and heat fluxes and to derive standard gas exchange coefficients. The method is used successfully at three shallow river sites where physical processes, especially heat exchange, are found to drive diurnal variations in gas exchange. The paper is well organized with careful, highly reasoned arguments for the approach and data treatments. The data examples are clear and mostly convincing.*

We thank the Referee for the positive and constructive overall evaluation.

*The only troubling part of the paper is sections that describe the possible methodological bias produced by temperature effects on the O2 sensor time series and how the authors have corrected their measurements for this bias. Although I agree this bias is likely and needs to be understood, I do not think the authors have shown they really know its magnitude. They estimate a ~3% change in the oxygen reading per 1 degree C, and even with relatively small temperature fluctuations (<0.1 oC) this creates a bias about 3 times the measured signal (Figure 6). What if the effect was 4% or 2% instead of 3%? How consistent is the effect between optode sensors and their films? Is the effect proportional to the oxygen concentration or independent of oxygen concentration? Since the response time of the thermistor is faster than the optode, does this alter the correction? In short, the authors need to independently measure the magnitude of the temperature effect before applying a correction. This might be done with experiments where the oxygen partial pressure is held constant but temperature varied. Otherwise, the applied corrections may be creating more bias than they are removing.*

Temperature effects or biases were an unexpected ancillary finding – the main focus is on quantifying air-water gas exchange by aquatic eddy covariance – but we can see that some important details were missing, or not explained adequately. The following points were raised by the referee:

1) *They estimate a ~3% change in the $O_2$ reading per $1^oC$.* – This is not an estimate, it is a calculation based on the signal conversion equations given by the dual $O_2$-temperature sensor's manufacture (JFE Advantech). According to the sensor manual we have (using manufacture's nomenclature):

$$P' = \frac{A}{1 + D(t - 25) + F(t - 25)^2} + \frac{B}{N\{1 + D(t - 25) + F(t - 25)^2\} + C}$$

$$P = G + H \times P'$$

where $A$, $B$, $C$, $D$, $F$, $G$, and $H$ are fitting constants determined by the manufacture, $t$ the temperature [°C], $N$ the instrument output for $O_2$ [0 – 5 Volt], and $P$ the dissolved $O_2$ concentration [%]. By varying $t$ for fixed values of $N$, the sensor's temperature coefficient (% change in $O_2$ concentration reading caused by a temperature change of 1 °C) was calculated to range between 2.7 – 3.4%. We did the lab experiments suggested by the Referee and measured a temperature coefficient of 2.9. We can add this extra information in Section 3.3 describing temperature effects.

2) *What if the effect was 4% or 2% instead of 3%?* – Assuming that the temperature fluctuations are the same, the temperature bias (false flux) is proportional with the temperature coefficient. We can add this information as well

3) *How consistent is the effect between optode sensors and their films?* – Because the coefficients $A$, $B$, $C$, $D$, $F$, $G$, and $H$ only change a small amount, if at all (< 0 – 10%), from sensor to sensor and from film to film, we conclude that the effect is well-described by our numbers provided above. We can, but suggest not to, add these very specific details to the text.

4) *Is the effect proportional to the $O_2$ concentration or independent of $O_2$ concentration?* – From the definition of the temperature coefficient (% change in $O_2$ concentration reading caused by a temperature change of 1 °C), the effect is proportional to the $O_2$ concentration. We can add this information to the text.

5) *Since the response time of the thermistor is faster than the optode, does this alter the correction?* – The response time curves presented in Berg et al. (2016) were determined separately for temperature and for $O_2$. In additional lab tests we inserted the dual $O_2$-temperature sensor from air into a water bath with both a significantly different temperature and $O_2$ concentration than the air. Since the response time curves looked similar to the ones shown in Berg et al. (2016), we assess that the minor difference in response times for temperature (0.34 s) and $O_2$ (0.51 s) does not affect the flux calculation or amplify the temperature bias. We can add this information, but suggest not to, as it appears to us to be too much detail. Again, the main focus of the manuscript is on quantifying air-water gas exchange by aquatic eddy covariance.

*If the authors can address the above concern any remaining revisions to the paper will be minor. Below are listed areas by line number that might be clarified.*

*Line 25. I question whether it is known that physical controls are "prevalent in lotic systems". Perhaps it would be better to say "can be prevalent in lotic systems and adds uncertainty to assessments of biological activity for such systems that are based on water column O2 concentration changes".*

We can change the text as recommended.

*Line 30. What is meant by "erosion in the surface water"? Erosion of what?*

We can clarify this by changing the sentence to: 'This was presumably caused by formation and erosion of vertical temperature-density gradients in the surface water driven by the heat flux into or out of the river that controlled the turbulent mixing'.

*Lines 78-80. Awkward sentence. Please restructure.*

We can change the sentence to: 'Turbulence, or turbulent-like motion, that affects or controls the thickness of the film on the water side, and thus the diffusive resistance to gas transport, can be driven by conditions both below and above the air-water interface'.

*Line 87. Omit "but" in this sentence.*

We can do so.

*Line 93. Indicate where and how the tracer additions are made.*

We can clarify this by changing the sentence to: "For smaller rivers and streams they include targeted parallel up-and across-stream additions of volatile tracers (e.g. propane) and hydrologic tracers (e.g. dissolved chloride), the latter is added to correct for dilution of propane due to hyporheic mixing (Genereux and Hemond 1992; Koopmans and Berg 2015)".

*Lines 103-104. Change "studied" to "studies" and then clarify what is meant by "fitting measurements done in other aquatic systems".*

We can clarify this by changing the sentence to: "Partly motivated by the substantial and often methodologically challenging effort required to measure $k$ at specific sites with any of these approaches, many studies have simply relied on general empirical correlations for $k$

produced by fitting *k* values measured for other similar aquatic systems (Raymond and Cole 2001; Borges et al. 2004; Cole et al. 2010)".

*Line 108. "many standard estimates" of what? Please clarify. Are you talking about carbon budgets?*

We can add: "...such as gross primary production, respiration, and net ecosystem metabolism".

*Line 125. Reword as: "we were able to derive parallel fluxes. . ."*

We can do so.

*Line 126. Reword as: "proof-of-concept tests that were up to 40 hours long at three river sites."*

We can do so.

*Line 131. Reword as: "All measurements were made from. . ."*

We can do so.

*Line 140. How was the measurement position ~5 cm below the interface determined?*

We can clarify this by changing the sentence to: "This type of ADV allowed the sensor head to be positioned facing upwards (Fig. 1) while recording the velocity field right below the air-water interface, typically ~4 cm. This distance was determined post deployment from standard ADV output".

*Later (line 222) can you indicate how sensitive the storage term correction is to changes in this measurement?*

We can add this information in Section 3.1 where we present the $O_2$ fluxes shown in Fig. 2.

*Line 144. Why is the senor not identified as manufactured by Rinko?*

We can add this information.

*Line 155. Separate into two sentences here. Indicate how reproducible the response times are with each fresh sensor film put on the optode tip.*

We can split the sentence in two. We can add this information in Section 2.1 where we first present the sensor's response times for $O_2$ concentration measurements.

*Line 158. Why reference Fig. 2a here?*

This is a mistake. We can correct it to Fig. 1a.

*Line 169. Change to "PAR sensor".*

We can do so.

*Line 183. "as level as possible" is vague. Can you indicate within a certain number of degrees from vertical? Please clarify how tilt changes were corrected for within the time span of a 15-min burst as the sensor must bob up and down some.*

We can reword the sentence to: "Using a level and by placing dive weights on the platform (Fig. 1b) care was taken to ensure that it was horizontal within the tolerance of the level to minimize post-processing rotations of the velocity field to correct for sensor tilt". We don't think it is possible to specify a value for this tilt. However, for the deployment shown in Figs. 2 and 3, rotations to nullify the mean vertical and transverse velocity, resulted in an average rotation angle with horizontal of only 1.3 degrees, and did not affect the flux calculation flux. We can add this information in Section 3.1 presenting this deployment.

Also, as described in Section 2.2, all river sites used for our proof-of-concept tests were chosen to have "smooth and quietly flowing water without standing riffles or waves". As a result, our sensors did not bob up and down during measurements. Consequently, a correction for such complex sensor movements was deemed unnecessary. We can add a sentence stating that in Section 2.2.

*Line 191. The key word here is "detectable". There may still be high frequency signals lost because they are not detectable by these sensors.*

We agree, and note that we did use the word "detectable" here. We doubt though, that the "undetectable" part of the flux signal has any significant magnitude given the steep drop-off of the flux contribution at the high-frequency end of the co-spectrum (Fig. 4) combined with the sensors response time ($t_{90\%}$: 0.51 s for $O_2$ and 0.34 s for temperature). We mention this at the very end of section 3.1 and suggest that we not add a more detailed discussion of this complex question.

*Line 227. It would be helpful here if the authors gave more information about how the "Spectra version 1.2" code treats the data. Also, what is meant by "several consecutive data segments"? How does this relate to what is shown in Fig. 4?*

We can add the requested information so that the paragraph reads: "To examine the eddy frequencies that carried the flux signal, cumulative co-spectra of the $O_2$ concentration and the vertical velocity were calculated for representative periods in each deployment with minimally varying fluxes using the software package Spectra version 1.2 (P. Berg unpubl.). This software performs essentially the identically flux calculation in the frequency domain after fast Fourier transforming the de-trended data as EddyFlux does in the time domain. Both software packages rely on the same means of de-trending and time shifting data".

Also, in the presentation of Fig. 4, the specific time interval behind the two co-spectra is mentioned specifically.

*Line 237. It would be helpful for the authors to present the relationships for calculating k600 even though they are in the papers cited.*

To do this in a meaningful way would add two equations and a separate paragraph to explain this calculation well. Since it is a standard conversion in the gas exchange literature we suggest that we do not add this. However, because the conversion is outlined best and most straight forward in the referenced Cole et al. 2010 paper, we suggest to remove the two other papers cited.

*Line 256. Since the data is presented as hourly fluxes, why not change the units in the figures to per hour rather than per day?*

We prefer to use the unit mmol m$^{-2}$ day$^{-1}$, in part because this unit is often used for measures such as net ecosystem metabolism.

*Line 269. Suggest reword as: "controlled by a driver apart from the river current velocity or winds…".*

We can do so.

*Line 272. Unclear what results are being referred to here.*

These are data from a stable independent dual $O_2$-temperature sensor. It is defined specifically in line 167, and we can specifically add/state that we refer to this sensor as "the independent sensor" throughout the manuscript.

*Line 317. I do not see why the authors reference Gundersen et al. 1998 here? This paper discusses the temperature sensitivity of oxygen microelectrodes that operate by different principles than optodes. The microelectrode temperature effect is usually related to the gas solubility in the membrane and changes in the diffusion rate.*

This referenced paper does indeed focus on microelectrodes. It is the only reference we have been able to find that gives information on the temperature coefficient for any type of fast-responding $O_2$ sensor. It is relevant because microelectrodes that apparently suffer from the same temperature dependency as optical sensors are still by far the most common sensor type used for aquatic eddy covariance. However, we acknowledge that this was not explained well, and we can elaborate on this as suggested in our response to the Referee's main comment (see above).

*Lines 352-355. Good argument here. Correct the spelling of "concentration" in line 355.*

We can correct this.

*Line 371. Reword as: "This, in turn, changed the. . .."*

We can correct this.

*Line 380-383. Can the authors take this argument further perhaps with an illustrative example?*

We do not have a measure for the average water depth, but we can assess a value based on visual inspection of the site. This would allow us to use the "open water" technique (Odum 1956) to estimate the benthic flux for comparison.

*Line 399. Do the authors have any temperature profiles from their sites that may illustrate temperature stratification during the day?*

No unfortunately not.

*Line 410. Change to: "was first developed"*

We can correct this.

*Lines 416-417. It is unsatisfying that the authors call for more studies of the temperature bias. As noted above, they need to include more concrete studies in the context of this paper.*

In our detailed response above to the Referee's main comment we suggest that we add more information along these lines. However, the main focus of our manuscript is the new approach for determining air-water gas exchange rates and coefficients, whereas the temperature bias is an ancillary finding. Conversely, we find it acceptable to suggest that more work is needed to further document this bias.

*Line 436-437. It would be helpful if earlier in the paper the authors indicated the magnitude of the O2 storage term relative to Jeddy (Equation 3).*

We can add this information in Section 3.1 where we present the $O_2$ fluxes shown in Fig. 2.

*Table 1. Add standard deviations to the parameters in the right three columns.*

We have reported SEs throughout the paper and can add these to Table 1.

*Figure 1c. Add arrows to indicate each identified item and indicate that the "independent dual O2-temperature sensors" are the miniDOT sensors and the sensor used for EC is a Rinko sensor.*

We can add this information.

---

## Author Comment (AC2) · 2 Nov 2017

**Response to Anonymous Referee #2:**

Based on comments from Referee #2 we plan to revise our manuscript as outlined below. All comments from the Referee are written with 'Italic' font. Our responses are written with 'normal' font.

*The manuscript by Peter Berg and Michael Pace investigates air-water gas exchange at three shallow river sites. The authors focus on the determination of oxygen and temperature exchange by using the eddy covariance technique on a floating platform to assess gas exchange coefficients. The major findings show that oxygen dynamics (on an hourly scale) are largely independent of current velocities and biological activity; instead oxygen dynamics are driven by heat exchange, i.e. changing oxygen saturations. Furthermore, the authors point out the importance of high resolution temperature measurements to correct for the oxygen sensor specific temperature sensitivity.*

*The manuscript is well written / structured and of interest for a broader readership as the results have important implications for the growing community that uses the aquatic eddy covariance technique. The approach to determine gas exchange coefficients using the temperature+oxygen eddy covariance technique is also a methodological advance.*

We thank the Referee for the positive and constructive overall evaluation.

*However, the manuscript is lacking some important details and it would benefit from a more in-depth analysis of the interesting and promising dataset. The heat-exchange driven oxygen fluxes are only masking the more interesting biogeochemical processes and are overemphasized. In the current version, the discussion about the temperature bias raises more questions than it actually resolves. See below for a detailed argumentation.*

*Argumentation:*

*1. One of the key findings and also a major part of the discussion is that heat exchange is driving most of the oxygen dynamics in shallow-water rivers. This is reasonable on timescales of hours, however, the physical process is only masking the biogeochemical processes which are still occurring and which are of importance. Based on the dataset it should be easily possible to distinguish between the biologically induced oxygen fluxes and the heat exchange induced oxygen fluxes. On the long run, the heat exchange induced fluxes should also average out implying a limited role for net exchange fluxes. When the authors follow my recommendation they could subtract the heat exchange induced oxygen flux from the total flux. I am convinced that this procedure will reveal good correlations with parameters like flow velocity and biological activity.*

We agree that physical processes are "*masking the biogeochemical processes which are still occurring and which are of importance*", but we do not understand how "*it should be easily possible to distinguish between the biologically induced oxygen fluxes and the heat exchange induced oxygen fluxes*". We agree that this would be desirable but we cannot see a way to split the total $O_2$ flux that we measure into these two components. It is suggested to "*subtract the heat exchange induced oxygen flux from the total flux*". How do we quantify the heat exchange induced $O_2$ flux? Maybe this suggestion is rooted in misreading Fig. 3 which shows the actual heat flux, and not the heat exchange induced $O_2$ flux?

*2. The effect of temperature fluctuations on the oxygen measurement is convincing but in the current version of the manuscript it raises several question that need to be addressed:*

Referee #1 echoed this point too, stating that important details on the effect of temperature fluctuations were missing, or not explained adequately. Please see our response to Referee #1 and the additional information below.

*Resolution: The authors are discussing the response time of the temperature sensor, which is indeed in the range of the oxygen sensor. However, the sensor tip is much thicker (8mm, line 154) which implies that the spatial resolution is limiting the minimum eddy size, i.e. frequency, that can be resolved.*

The diameters of the thermistor and the active $O_2$ sensing foil are ~1 mm and ~5 mm, respectively, and the thermistor is positioned ~2 mm away from the edge of the foil. These dimensions should be considered in relation to the measuring volume of the Acoustic Doppler Velocimeter (ADV) which has a 14 mm diameter and is 14 mm tall. In that light, the limiting factor of what eddy sizes, or frequencies, can be resolved is associated with the ADV and not the dual $O_2$-temperature sensor. We can explain this point in more detail.

*Sampling Rate / Correction Procedure: The "real" sampling rate of the oxygen sensor and temperature sensor differ as the response times are slightly different and there is also a distance between the two sensors. How did the authors ensure that the temperature measured is similar to the one at the oxygen sensor tip? Did the authors also apply a time shift correction?*

The response times reported in Berg et al. (2016) are 0.34 s for temperature and 0.51 s for $O_2$. Because of that, and because of the slightly different distances from the thermistor and $O_2$ sensing foil to the center of the ADV's measuring volume, we applied independent time shift corrections for the heat flux and the $O_2$ flux. We did explain how the time shift was performed for $O_2$, and suggest that we add in our revised manuscript that the correction was done independently for heat.

*Range of Error: The example depicted in Figure 6 indicates that in the case of systems with large heat exchange, basically all measurements without temperature-correction are wrong. Therefore, this kind of correction needs a careful assessment. It would be interesting to see the temperature correction applied in Figure 2 for the hourly oxygen fluxes.*

Again, Referee #1 echoed this point. Please see our response to Referee #1 and the additional information above and below. All $O_2$ fluxes we report, except some of those in Fig. 6, are temperature corrected. We can state this stronger in Section 2.3 where we describe our flux calculation protocol. Also, as suggested we can add non temperature corrected $O_2$ fluxes (as small black dots on top of red bars) in Fig. 2 to illustrate the importance of this correction.

*Figure / table / line specific comments:*

*Figure 5: It is of interest to present the missing correlation between the gas exchange coefficient and flow velocity, however this should be contrasted by an existing correlation. An example could be the comparison of the temperature gradient versus the gas exchange coefficient. This correlation would strengthen the argumentation.*

We do not understand the first suggestion here. With respect to showing the gas exchange coefficient vs. the temperature gradient (the vertical one?), unfortunately we do not have any temperature measurements down through the very top of the water column, but this would indeed be an interesting analysis to make in future studies.

*Table 1: Most of the oxygen flux is driven by heat exchange, which shows most of its variation on a daily basis. The presented oxygen fluxes are averaged in time intervals of 1 hour – 12 hours and are, therefore, strongly biased. As a result the variability within the oxygen fluxes is arbitrary as it only depends on the cut-off time.*

We disagree that our tabulated $O_2$ fluxes are "strongly biased" and that their variability is "arbitrary" and "only depends on the cut-off time". In Section 3.2 we stated specifically that the fluxes in Table 1 and the derived gas exchange coefficients represent periods of time with several successive 15-min time intervals that had little variation and appeared to represent a particular field condition. We don't know how to address this point any better.

*Line 103: many studies.*

We can fix this typo.

*Line 135: It is not very convincing that a floating platform is stable when fixed as described. Actually, I would expect movements that are in the range of the eddies that carry the oxygen signal.*

As we state in Section 2.2, all sites were picked to have a smooth and quietly flowing water without standing riffles or waves. We regard our tests as proof-of-concept deployments and aimed carefully at keeping the field conditions as simple as possible. We didn't observe any vertical movements of the platform, or even eddies distorting the air-water interface. Although less critical for the flux calculation, we did not observe any lateral movements of the platform either due to the two-point anchoring system we used. We can expand the explanation of this in Section 2.1.

*Line 185 and 196: How accurate was the positioning / how big was the sensor tilt? It is not clear if the correction for the sensor tilt was performed or not.*

It is difficult to put a number on this sensor tilt, but for the deployment featured in Figs. 2 and 3, the average rotations with horizontal to nullify the mean vertical and transverse velocity was only 1.3 degrees and did not affect the flux calculation flux. We can add this information in Section 3.1 where we present this deployment.

*Line 210: It should be stated in which range the time shift is. Considering the very constant flow velocity and the known response times it should be possible to calculate it. The time shift should not be bigger than the time it needs to travel from the ADV measuring volume to the sensor tip + response time!?*

Due to a micro boundary layer forming on the $O_2$ sensing foil, the actual time shift found as described in Section 2.3 is slightly larger than suggested by the Referee. Again, for the deployment in Fig. 2, the averaged time shift equaled 1.3 s. We can add this information in Section 3.1 where we present this deployment.

*Line 219: Equation 3 is not adequately described, what does the second term imply, how is it measured, what is the range relative to the eddy covariance flux.*

We agree, Eq. 3 was not adequately explained. We can correct this and also provide an average number for the magnitude of the storage relative to the eddy flux as suggest by Referee #1.

*Line 239: To my knowledge "lumped" is not a statistical method.*

Please note that we do not claim that. We find that the description of how to generate 8 Hz data from 64 Hz data is sufficient.

---

## Author Response (AR1)

Dear Jack (Associate Editor Jack Middelburg),

Thank you for your positive response to our BGD submission. As requested, we have detailed all relevant changes that we have made to the manuscript below. The changes closely follow those we proposed in our on-line response to the two Referees with one exception: we did not include the application of the open-water method (Odum, 1956) that we suggested in response to one of Referee #1's comments. Our rational for not including this is given below.

All original comments from the Referees are written with 'Italic' font. How we have changed the texts is described with 'normal' font.

**Changes made based on comments from Anonymous Referee #1:**

This manuscript describes an important methodological advance for aquatic sciences by demonstrating that the eddy covariance method can be applied from the water-side of the airwater interface to measure oxygen and heat fluxes and to derive standard gas exchange coefficients. The method is used successfully at three shallow river sites where physical processes, especially heat exchange, are found to drive diurnal variations in gas exchange. The paper is well organized with careful, highly reasoned arguments for the approach and data treatments. The data examples are clear and mostly convincing.

We thank the Referee for the positive and constructive overall evaluation.

The only troubling part of the paper is sections that describe the possible methodological bias produced by temperature effects on the O2 sensor time series and how the authors have corrected their measurements for this bias. Although I agree this bias is likely and needs to be understood, I do not think the authors have shown they really know its magnitude. They estimate a  $\sim 3\%$  change in the oxygen reading per 1 degree C, and even with relatively small temperature fluctuations (<0.1 oC) this creates a bias about 3 times the measured signal (Figure 6). What if the effect was 4% or 2% instead of 3%? How consistent is the effect between optode sensors and their films? Is the effect proportional to the oxygen concentration or independent of oxygen concentration? Since the response time of the thermistor is faster than the optode, does this alter the correction? In short, the authors need to independently measure the magnitude of the temperature effect before applying a correction. This might be done with experiments where the oxygen partial pressure is held constant but temperature varied. Otherwise, the applied corrections may be creating more bias than they are removing.

Temperature effects or biases were an unexpected ancillary finding – the main focus was on quantifying air-water gas exchange by aquatic eddy covariance – but we can see that some

important details were missing, or not explained adequately. The following points were raised by the referee:

1) They estimate a  $\sim 3\%$  change in the  $O_2$  reading per  $1^\circ C$ . – This is not an estimate, it is a calculation based on the signal conversion equations given by the dual  $O_2$ -temperature sensor's manufacture (JFE Advantech). According to the sensor manual we have (using manufacture's nomenclature):

$$P' = \frac{A}{1 + D(t - 25) + F(t - 25)^2} + \frac{B}{N\{1 + D(t - 25) + F(t - 25)^2\} + C}$$

$$P = G + H \times P'$$

where *A*, *B*, *C*, *D*, *F*, *G*, and *H* are fitting constants determined by the manufacture, *t* is the temperature [°C], *N* is the instrument output for  $O_2$  [0 – 5 Volt], and *P* is the dissolved  $O_2$  concentration [%]. By varying *t* for fixed values of *N*, the sensor's temperature coefficient (% change in  $O_2$  concentration reading caused by a temperature change of 1 °C) was calculated to range between 2.7 – 3.4%.

We did the lab experiments suggested by the Referee and measured a temperature coefficient of 2.9. We have added this extra information in Section 3.3 describing temperature effects.

- 2) What if the effect was 4% or 2% instead of 3%? Assuming that the temperature fluctuations are the same, the temperature bias (false flux) is proportional with the temperature coefficient. We have added this information as well.
- 3) *How consistent is the effect between optode sensors and their films?* Because the coefficients *A*, *B*, *C*, *D*, *F*, *G*, and *H* only change a small amount, if at all (< 0 10%), from sensor to sensor and from film to film, we conclude that the effect is well-described by the information we have added as outlined above. We have chosen not to add these very specific details.
- 4) Is the effect proportional to the O2 concentration or independent of O2 concentration? From the definition of the temperature coefficient (% change in O2 concentration reading caused by a temperature change of 1 °C), the effect is proportional to the O2 concentration. We have added this information.
- 5) Since the response time of the thermistor is faster than the optode, does this alter the *correction?* The response time curves presented in Berg et al. (2016) were determined separately for temperature and for O2. In additional lab tests we inserted

the dual  $O_2$ -temperature sensor from air into a water bath with both a significantly different temperature and  $O_2$  concentration than the air. Since the response time curves looked similar to the ones shown in Berg et al. (2016), we assess that the minor difference in response times for temperature (0.34 s) and  $O_2$  (0.51 s) does not affect the flux calculation or amplify the temperature bias. We have added this information.

*If the authors can address the above concern any remaining revisions to the paper will be minor. Below are listed areas by line number that might be clarified.*

Line 25. I question whether it is known that physical controls are "prevalent in lotic systems". Perhaps it would be better to say "can be prevalent in lotic systems and adds uncertainty to assessments of biological activity for such systems that are based on water column O2 concentration changes".

We have changed the sentence to: "This physical control of gas exchange can be prevalent in lotic systems and adds uncertainty to assessments of biological activity that are based on measured water column  $O_2$  concentration changes".

**Line 30. What is meant by "erosion in the surface water"? Erosion of what?**

We have clarify this by changing the sentence to: "This was presumably caused by the formation and erosion of vertical temperature-density gradients in the surface water driven by the heat flux into or out of the river that affected the turbulent mixing".

**Lines 78-80. Awkward sentence. Please restructure.**

We have changed the sentence to: "Turbulence, or turbulent-like motion, that affects or controls the thickness of the film on the water side, and thus the diffusive resistance to gas transport, can be driven by conditions both below and above the air-water interface".

**Line 87. Omit "but" in this sentence.**

We have done so.

*Line 93. Indicate where and how the tracer additions are made.*

We have clarified this by changing the sentence to: "For smaller rivers and streams they include targeted parallel up-and across-stream additions of volatile tracers (e.g. propane) and hydrologic tracers (e.g. dissolved chloride), where the latter is added to correct for

dilution of propane due to hyporheic mixing (Genereux and Hemond 1992; Koopmans and Berg 2015)".

Lines 103-104. Change "studied" to "studies" and then clarify what is meant by "fitting measurements done in other aquatic systems".

We have corrected the typo and addressed this question by changing the sentence to: "Partly motivated by the substantial and often methodologically challenging effort required to measure k at specific sites with any of these approaches, many studies have simply relied on general empirical correlations for k produced by fitting k values measured for other similar aquatic systems (Raymond and Cole 2001; Borges et al. 2004; Cole et al. 2010)".

*Line 108. "many standard estimates" of what? Please clarify. Are you talking about carbon budgets?*

We have added: "...such as gross primary production, respiration, and net ecosystem metabolism".

*Line 125. Reword as: "we were able to derive parallel fluxes. . ."*

We have done so.

*Line 126. Reword as: "proof-of-concept tests that were up to 40 hours long at three river sites."*

We have done so.

Line 131. Reword as: "All measurements were made from. . ."

We have done so.

*Line 140. How was the measurement position*  $\sim$  5 *cm below the interface determined?*

We have clarified this by changing the sentence to: "This type of ADV allowed the sensor head to be positioned facing upwards (Fig. 1) while recording the velocity field right below the air-water interface, typically  $\sim$ 4 cm. This distance was determined from standard ADV output".

Later (line 222) can you indicate how sensitive the storage term correction is to changes in this measurement?

We have added this information in Section 3.1 where we present the  $O_2$  fluxes shown in Fig. 2.

Line 144. Why is the senor not identified as manufactured by Rinko?

We have added this information.

*Line 155. Separate into two sentences here. Indicate how reproducible the response times are with each fresh sensor film put on the optode tip.*

We have split the sentence in two. We have added this information in Section 2.1 where we first present the sensor's response times for  $O_2$  concentration measurements.

Line 158. Why reference Fig. 2a here?

This was a mistake. We have corrected it to Fig. 1a.

Line 169. Change to "PAR sensor".

We have done so.

Line 183. "as level as possible" is vague. Can you indicate within a certain number of degrees from vertical? Please clarify how tilt changes were corrected for within the time span of a 15-min burst as the sensor must bob up and down some.

We have reworded the sentence to: "Using a level and by placing dive weights on the platform (Fig. 1b) care was taken to ensure that it was horizontal within the tolerance of the level to minimize post-processing rotations of the velocity field to correct for sensor tilt".

We don't think it is possible to specify a value for this tilt. However, for the deployment shown in Figs. 2 and 3, rotations to nullify the mean vertical and transverse velocity, resulted in an average rotation angle with horizontal of only 1.3 degrees, and did not affect the flux calculation. We added this information in Section 3.1 where this deployment is presented.

Also, as described in Section 2.2, all river sites used for our proof-of-concept tests were chosen to have "smooth and quietly flowing water without standing riffles or waves". As a result, our sensors did not bob up and down during measurements. Consequently, a correction for such complex sensor movements was deemed unnecessary. We added a sentence stating that in Section 2.2.

Line 191. The key word here is "detectable". There may still be high frequency signals lost because they are not detectable by these sensors.

We agree, and note that we did use the word "detectable" here. We doubt though, that the "undetectable" part of the flux signal has any significant magnitude given the steep drop-off of the flux contribution at the high-frequency end of the co-spectrum (Fig. 4) combined with the sensors response time ( $t_{90\%}$ : 0.51 s for O2 and 0.34 s for temperature). We mention this at the very end of section 3.1 and have not added a more detailed discussion of this complex question.

Line 227. It would be helpful here if the authors gave more information about how the "Spectra version 1.2" code treats the data. Also, what is meant by "several consecutive data segments"? How does this relate to what is shown in Fig. 4?

We have added the requested information so that the paragraph reads: "To examine the eddy frequencies that carried the flux signal, cumulative co-spectra of the O2 concentration and the vertical velocity were calculated for representative periods in each deployment with minimally varying fluxes using the software package Spectra version 1.2 (P. Berg unpubl.). This software essentially performs the identical flux calculation in the frequency domain after fast Fourier transforming the de-trended data as EddyFlux does in the time domain. Both software packages rely on the same means of de-trending and time shifting data".

Also, in the presentation of Fig. 4, the specific time interval behind the two co-spectra is mentioned specifically.

Line 237. It would be helpful for the authors to present the relationships for calculating k600 even though they are in the papers cited.

To do this in a meaningful way would add two equations and a separate paragraph to explain this calculation well. Since it is a standard conversion in the gas exchange literature we suggest that we do not add this. However, because the conversion is outlined best and most straight forward in the referenced Cole et al. 2010 paper, we have removed the two other papers cited.

Line 256. Since the data is presented as hourly fluxes, why not change the units in the figures to per hour rather than per day?

We prefer to use the unit mmol m-2 day-1, in part because this unit is often used for measures such as net ecosystem metabolism.

*Line 269. Suggest reword as: "controlled by a driver apart from the river current velocity or winds...".*

We have done so.

*Line 272. Unclear what results are being referred to here.*

These are data from a stable independent dual  $O_2$ -temperature sensor. It is defined specifically in line 167, and we have added that we refer to this sensor as "the independent sensor" throughout the manuscript.

Line 317. I do not see why the authors reference Gundersen et al. 1998 here? This paper discusses the temperature sensitivity of oxygen microelectrodes that operate by different principles than optodes. The microelectrode temperature effect is usually related to the gas solubility in the membrane and changes in the diffusion rate.

This referenced paper does indeed focus on microelectrodes. It is the only reference we have been able to find that gives information on the temperature coefficient for any type of fastresponding O2 sensor. It is relevant because microelectrodes that apparently suffer from the same temperature dependency as optical sensors are still by far the most common sensor type used for aquatic eddy covariance. However, we acknowledge that this was not explained well, and we have elaborated on this as suggested in our response to the Referee's main comment (see above).

*Lines 352-355. Good argument here. Correct the spelling of "concentration" in line 355.*

We have corrected this.

Line 371. Reword as: "This, in turn, changed the. . .."

We have corrected this.

Line 380-383. Can the authors take this argument further perhaps with an illustrative example?

If we had a good measurement of the average water depth, or a way to asses it, we could apply the standard "open water" technique (Odum, 1956) and estimate the benthic flux for evaluation. However, without this information we think this exercise would become too speculative and uncertain, and thus, not serve as a meaningful example supporting our point.

*Line 399. Do the authors have any temperature profiles from their sites that may illustrate temperature stratification during the day?*

No, unfortunately not, but this is something we would like to add in future studies.

Line 410. Change to: "was first developed"

We have corrected this.

*Lines 416-417. It is unsatisfying that the authors call for more studies of the temperature bias. As noted above, they need to include more concrete studies in the context of this paper.*

In response to the Referee's main comment, we have added more information along these lines (see above). However, the main focus of our manuscript is the new approach for determining air-water gas exchange rates and coefficients, whereas the temperature bias is an ancillary finding. Conversely, we find it acceptable to suggest that more work is needed along those lines.

*Line 436-437. It would be helpful if earlier in the paper the authors indicated the magnitude of the 02 storage term relative to Jeddy (Equation 3).*

We have added this information in Section 3.1 where we present the  $O_2$  fluxes shown in Fig. 2.

Table 1. Add standard deviations to the parameters in the right three columns.

We have reported SEs throughout the paper and can add these to Table 1.

Figure 1c. Add arrows to indicate each identified item and indicate that the "independent dual O2-temperature sensors" are the miniDOT sensors and the sensor used for EC is a Rinko sensor.

We have added this information.

**Changes made based on comments from Anonymous Referee #2:**

The manuscript by Peter Berg and Michael Pace investigates air-water gas exchange at three shallow river sites. The authors focus on the determination of oxygen and temperature exchange by using the eddy covariance technique on a floating platform to assess gas exchange coefficients. The major findings show that oxygen dynamics (on an hourly scale) are largely independent of current velocities and biological activity; instead oxygen dynamics are driven by heat exchange, i.e. changing oxygen saturations. Furthermore, the authors point out the importance of high resolution temperature measurements to correct for the oxygen sensor specific temperature sensitivity.

The manuscript is well written / structured and of interest for a broader readership as the results have important implications for the growing community that uses the aquatic eddy covariance technique. The approach to determine gas exchange coefficients using the temperature+oxygen eddy covariance technique is also a methodological advance.

We thank the Referee for the positive and constructive overall evaluation.

However, the manuscript is lacking some important details and it would benefit from a more in-depth analysis of the interesting and promising dataset. The heat-exchange driven oxygen fluxes are only masking the more interesting biogeochemical processes and are overemphasized. In the current version, the discussion about the temperature bias raises more questions than it actually resolves. See below for a detailed argumentation.

**Argumentation:**

1. One of the key findings and also a major part of the discussion is that heat exchange is driving most of the oxygen dynamics in shallow-water rivers. This is reasonable on timescales of hours, however, the physical process is only masking the biogeochemical processes which are still occurring and which are of importance. Based on the dataset it should be easily possible to distinguish between the biologically induced oxygen fluxes and the heat exchange induced oxygen fluxes. On the long run, the heat exchange induced fluxes should also average out implying a limited role for net exchange fluxes. When the authors follow my recommendation they could subtract the heat exchange induced oxygen flux from the total flux. I am convinced that this procedure will reveal good correlations with parameters like flow velocity and biological activity.

We agree that physical processes are "masking the biogeochemical processes which are still occurring and which are of importance", but we do not understand how "it should be easily possible to distinguish between the biologically induced oxygen fluxes and the heat exchange induced oxygen fluxes". We agree that this would be desirable, but we cannot see a way to split the total  $O_2$  flux that we measured into these two components. It is suggested to "subtract the heat exchange induced oxygen flux from the total flux", but how do we quantify the heat exchange induced  $O_2$  flux? Maybe this suggestion is rooted in misreading Fig. 3 which shows the actual heat flux, and not the heat exchange induced  $O_2$  flux?

2. The effect of temperature fluctuations on the oxygen measurement is convincing but in the current version of the manuscript it raises several question that need to be addressed:

Referee #1 echoed this point too, stating that important details on the effect of temperature fluctuations were missing, or not explained adequately. Please see our response to Referee #1 and the additional information below.

Resolution: The authors are discussing the response time of the temperature sensor, which is indeed in the range of the oxygen sensor. However, the sensor tip is much thicker (8mm, line 154) which implies that the spatial resolution is limiting the minimum eddy size, i.e. frequency, that can be resolved.

The diameters of the thermistor and the active  $O_2$  sensing foil are ~1 mm and ~5 mm, respectively, and the thermistor is positioned ~2 mm away from the edge of the foil. These dimensions should be considered in relation to the measuring volume of the Acoustic Doppler Velocimeter (ADV) which has a 14 mm diameter and is 14 mm tall. In that light, the limiting factor of what eddy sizes, or frequencies, can be resolved is associated with the ADV and not the dual  $O_2$ -temperature sensor. We have explained this point in Section 2.1 where we describe our sensors.

Sampling Rate / Correction Procedure: The "real" sampling rate of the oxygen sensor and temperature sensor differ as the response times are slightly different and there is also a distance between the two sensors. How did the authors ensure that the temperature measured is similar to the one at the oxygen sensor tip? Did the authors also apply a time shift correction?

The response times reported in Berg et al. (2016) are 0.34 s for temperature and 0.51 s for  $O_2$ . Because of that, and because of the slightly different distances from the thermistor and  $O_2$  sensing foil to the center of the ADV's measuring volume, we applied independent time shift corrections for the heat flux and the  $O_2$  flux. We did originally explain how the time shift was performed for  $O_2$ , and we have added that this correction was applied independently for the heat flux.

Range of Error: The example depicted in Figure 6 indicates that in the case of systems with large heat exchange, basically all measurements without temperature-correction are wrong. Therefore, this kind of correction needs a careful assessment. It would be interesting to see the temperature correction applied in Figure 2 for the hourly oxygen fluxes.

The text was unclear about this, but the temperature correction was applied to the  $O_2$  fluxes shown in Fig. 2, as it was to all data we report. It is only for one data example shown in Fig. 6 that this correction was not applied to illustrate the severe effect that omitting the

temperature correction can have. We have stated this clearly in Section 2.3 where we describe our flux calculation protocol and also in the Fig. 6 legend.

*Figure / table / line specific comments:*

Figure 5: It is of interest to present the missing correlation between the gas exchange coefficient and flow velocity, however this should be contrasted by an existing correlation. An example could be the comparison of the temperature gradient versus the gas exchange coefficient. This correlation would strengthen the argumentation.

We do not understand the first suggestion here. With respect to showing the gas exchange coefficient vs. the temperature gradient (the vertical one?), unfortunately we do not have any temperature measurements down through the top of the water column, but this would indeed be an interesting analysis to make in future studies.

Table 1: Most of the oxygen flux is driven by heat exchange, which shows most of its variation on a daily basis. The presented oxygen fluxes are averaged in time intervals of 1 hour – 12 hours and are, therefore, strongly biased. As a result the variability within the oxygen fluxes is arbitrary as it only depends on the cut-off time.

We disagree that our tabulated O2 fluxes are "strongly biased" and that their variability is "arbitrary" and "only depends on the cut-off time". In Section 3.2 we stated specifically that the fluxes in Table 1 and the derived gas exchange coefficients represent periods of time with several successive 15-min time intervals that had little variation and appeared to represent a particular field condition. We don't know how to address this point any better.

Line 103: many studies.

We have fixed this typo.

Line 135: It is not very convincing that a floating platform is stable when fixed as described. Actually, I would expect movements that are in the range of the eddies that carry the oxygen signal.

As we state in Section 2.2, all sites were picked because they had smooth and quietly flowing water without standing riffles or waves. We regard our tests as proof-of-concept deployments and aimed carefully at keeping the field conditions as simple as possible. We didn't observe any vertical movements of the platform or even eddies distorting the airwater interface. Although less critical for the flux calculation, we did not observe any lateral

movements of the platform either due to the two-point anchoring system we used. We have expanded the explanation of this in Section 2.2.

Line 185 and 196: How accurate was the positioning / how big was the sensor tilt? It is not clear if the correction for the sensor tilt was performed or not.

It is difficult to put a number on this sensor tilt, but for the deployment featured in Figs. 2 and 3, the average rotation with horizontal direction to nullify the mean vertical and transverse velocity was only 1.3 degrees and did not affect the flux calculation. We have added this information in Section 3.1 where we present this deployment.

Line 210: It should be stated in which range the time shift is. Considering the very constant flow velocity and the known response times it should be possible to calculate it. The time shift should not be bigger than the time it needs to travel from the ADV measuring volume to the sensor tip + response time!?

Due to a micro boundary layer forming on the  $O_2$  sensing foil, the actual time shift found as described in Section 2.3 is slightly larger than suggested by the Referee. Again, for the deployment in Fig. 2, the averaged time shift equaled 1.3 s. We have added this information in Section 3.1 where we present this deployment.

*Line 219: Equation 3 is not adequately described, what does the second term imply, how is it measured, what is the range relative to the eddy covariance flux.*

We agree, Eq. 3 was not adequately explained. We have corrected this and also provided an average number for the magnitude of the storage relative to the eddy flux as suggest by Referee #1.

*Line 239: To my knowledge "lumped" is not a statistical method.*

Please note that we do not claim that. We find that the description of how to generate 8 Hz data from 64 Hz data is sufficient.

[revised manuscript text omitted]
 variationschanges, and thus they will give variable O2 readings at the same molar O2 concentration if the temperature changesvaries. Typical temperature coefficients (% change in O2 concentration reading caused by a temperature change of 1 °C) for Clark-type microelectrodes, still the most common sensor type used for aquatic eddy covariance, have values of ~3 %
- 365 (Gundersen et al. 1998). Lab measurements in which the O2 concentration was held constant but temperature varied showed that the fast-responding dual O2-temperature sensor used in this study has a temperature coefficient of 2.9 % if temperature correction was omitted. This characteristic of fast-responding O2 sensors implies that rapid temperature fluctuations associated with a turbulent heat flux will mistakenly be recorded as fluctuations in O2
- 370 concentration and bias the eddy flux calculation unless an instantaneous temperature correction of the O2 reading signal is performed. In this study, this correction was done using the we relied on a new fast-responding dual O2-temperature sensor's (Berg et al. 2016) which puts out rapid simultaneous readings of both the O2 concentration and the temperature reading from within a distance of a few mm of the O2 sensing foil 
[revised manuscript text omitted]